# Immune profiling-based targeting of pathogenic T cells with ustekinumab in ANCA-associated glomerulonephritis

Jonas Engesser[1,2,7], Robin Khatri[2,3,7], Darius P. Schaub[2,3,7], Yu Zhao[1,2,3], Hans-Joachim Paust[1,2], Zeba Sultana[1,2,3], Nariaki Asada[1,2], Jan-Hendrik Riedel[1,2], Varshi Sivayoganathan[1,2], Anett Peters[1], Anna Kaffke[1], Saskia-Larissa Jauch-Speer[1], Thiago Goldbeck-Strieder[1], Victor G. Puelles[1,4], Ulrich O. Wenzel[1], Oliver M. Steinmetz[1], Elion Hoxha[1,4], Jan-Eric Turner[1,2], Hans-Willi Mittrücker[2,5], Thorsten Wiech[4,6], Tobias B. Huber[1,2,4], Stefan Bonn[2,3,4,8]✉, Christian F. Krebs[1,2,4,8]✉ & Ulf Panzer[1,2,4,8]✉

Antineutrophil cytoplasmic antibody (ANCA)−associated vasculitis is a life-threatening autoimmune disease that often results in kidney failure caused by crescentic glomerulonephritis (GN). To date, treatment of most patients with ANCA-GN relies on non-specific immunosuppressive agents, which may have serious adverse effects and be only partially effective. Here, using spatial and single-cell transcriptome analysis, we characterize inflammatory niches in kidney samples from 34 patients with ANCA-GN and identify proinflammatory, cytokine-producing CD4[+] and CD8[+] T cells as a pathogenic signature. We then utilize these transcriptomic profiles for digital pharmacology and identify ustekinumab, a monoclonal antibody targeting IL-12 and IL-23, as the strongest therapeutic drug to use. Moreover, four patients with relapsing ANCA-GN are treated with ustekinumab in combination with low-dose cyclophosphamide and steroids, with ustekinumab given subcutaneously (90 mg) at weeks 0, 4, 12, and 24. Patients are followed up for 26 weeks to find this treatment well-tolerated and inducing clinical responses, including improved kidney function and Birmingham Vasculitis Activity Score, in all ANCA-GN patients. Our findings thus suggest that targeting of pathogenic T cells in ANCA-GN patients with ustekinumab might represent a potential approach and warrants further investigation in clinical trials.

Antineutrophil cytoplasmic antibody (ANCA)-associated vasculitis is a group of systemic autoimmune diseases characterized by inflamed and necrotic small to medium-sized blood vessels[1]. Kidney involvement is common and is associated with a substantial risk of end-stage renal disease and death. Renal involvement typically manifests as rapidly progressive crescentic glomerulonephritis (ANCA-GN) with a fast decline in kidney function[2–4]. Despite recent advances in the treatment and management of ANCA-GN, such as

[1]Department of Medicine III, University Medical Center Hamburg-Eppendorf, Hamburg, Germany. [2]Hamburg Center for Translational Immunology, University Medical Center Hamburg-Eppendorf, Hamburg, Germany. [3]Institute of Medical Systems Biology, Center for Biomedical AI, Center for Molecular Neurobiology Hamburg, Hamburg, Germany. [4]Hamburg Center for Kidney Health (HCKH), University Medical Center Hamburg-Eppendorf, Hamburg, Germany. [5]Institute for Immunology, University Medical Center Hamburg-Eppendorf, Hamburg, Germany. [6]Institute of Pathology, Division of Nephropathology, University Medical Center Hamburg-Eppendorf, Hamburg, Germany. [7]These authors contributed equally: Jonas Engesser, Robin Khatri, Darius P. Schaub. [8]These authors jointly supervised this work: Stefan Bonn, Christian F. Krebs, Ulf Panzer. ✉e-mail: s.bonn@uke.de; c.krebs@uke.de; panzer@uke.de

B cell depletion using rituximab[5–7] and complement C5a receptor blockade with avacopan[8], the rate of end-stage kidney disease and side effects remains high, emphasizing the unmet need for more effective and immunopathogenesis-based treatment strategies in ANCA-GN.

Several studies investigated the gene expression profiles of blood samples from patients with ANCA-associated vasculitis (AAV) showing distinct endotypes and potential prognostic biomarkers[9–11]. Moreover, recent flow cytometric, single-cell RNA sequencing (scRNA-seq), and immunohistochemical analyses of kidney biopsy samples have provided deeper insights into the pathological mechanisms mediated by immune cells in ANCA-GN[12–15]. However, the relevant specific spatial localization of immune cells and their cellular interactions are largely unknown. Decoding the localization and function of immune cells in the kidney is highly relevant because the local immune responses could be the drivers of renal injury and disease progression, offering unique opportunities for the identification and characterization of treatment targets for ANCA-GN.

ANCA-GN patients remain at significant risk of renal failure and increased mortality, highlighting the need to develop more effective and safer therapies. Here, we combine spatial transcriptomics and single-cell RNAseq and identify Th1 and Th17 cells as major contributors to immune-mediated renal injury in ANCA-GN. Based on these results we treat four ANCA-GN patients with ustekinumab, which specifically targets Th1 and Th17 cells, as add-on therapy. The rapid clinical response in all four patients suggests that ustekinumab could be a promising therapy and should be further investigated in clinical trials. Our approach to combining high-dimensional single-cell and spatial immune profiling with clinical and histopathological data facilitates personalized pathogenesis-based treatments and could be a promising strategy for other autoimmune diseases.

## Results

### Study cohort and experimental overview

We included two independent patient groups with biopsy-confirmed ANCA-GN from the Hamburg GN Registry[14,16] in our study (Fig. 1). The exploratory group consists of 34 ANCA-GN patients. From each of these patients, two renal biopsy cores were taken. One was used for routine pathological evaluation and the other sample was used for spatial ($n = 28$) and single-cell ($n = 27$) transcriptomic analysis (Fig. 1 and Table 1). The treatment group consists of four patients with relapsing ANCA-GN that were treated with ustekinumab, steroids, and low-dose cyclophosphamide and underwent single cell, and flow cytometry immune profiling, as well as pathological examination and clinical follow-up analysis for 26 weeks (Fig. 1).

### Spatial transcriptomics reveals inflammatory glomerular and tubulointerstitial niches linked with T cell activation in ANCA-GN

Kidney inflammation is a hallmark of ANCA-GN but the underlying immunopathology is not well understood. To characterize the inflammatory niches, pathological cell-cell interactions, and key molecular pathways that drive kidney inflammation in ANCA-GN, we generated spatial transcriptome (ST) sequencing data from 28 renal biopsies of the exploratory cohort using the Visium platform (Fig. 2a and Supplementary Data 1). By unsupervised clustering of the spatial data, we were able to define 12 tissue compartments. Based on marker gene expression, we identified normal glomeruli, inflamed glomeruli, tubulointerstitium, inflamed tubulointerstitium, vasculature, and several tubular compartments. The latter includes proximal tubules (PT), connecting tubules (CNT), distal convoluted tubules (DCT), collecting duct (CD), and loop of Henle (LOH) (Fig. 2b,c and Supplementary Fig. 1a–c)[17]. We verified our clustering-based annotations by comparison to expert annotations of glomerular compartments on H&E-stained images, exhibiting an annotation concordance of over 90% on normal glomerular regions (Supplementary Fig. 1d). To understand the compositional difference of the identified compartments between ANCA-GN and healthy controls, we included 8 healthy control samples in our analysis (Supplementary Fig. 1e). ANCA-GN samples were enriched for inflamed glomeruli and inflamed tubulointerstitium as compared to healthy control samples (Fig. 2c and Supplementary Data 2).

Next, we aimed to identify key molecular pathways and cell subtypes involved in immunopathology in the inflammatory niches of the kidney. An unsupervised analysis of cluster-defining genes from the two inflamed compartments identified T cell activation as the most differentially expressed gene ontology term (Fig. 2d and Supplementary Data 3–5). By co-analyzing the neighborhood composition of compartments and their enrichment for T cell-specific pathways, we found that gene sets for Th1 and Th17 cell differentiation as well as T cell-mediated cytotoxicity were enriched in inflamed compartments (Fig. 2e, f and Supplementary Data 4, 5). Further gene sets upregulated in inflamed glomerular compartments indicate increased interleukin-1, extracellular matrix organization, and regulation of fibroblast proliferation (Supplementary Data 5).

### Enrichment of proinflammatory cytokine-producing Th1/Tc1 and Th17/Tc17-like effector T cells in the kidney of ANCA-GN patients

To further clarify and define the role of specific T cell subtypes and their signaling cascades in ANCA-GN, we generated a single cell transcriptome and epitope atlas of T cells, encompassing 72,416 T cells from renal biopsy and blood samples of 27 ANCA-GN patients of the exploratory cohort (Fig. 3a and Supplementary Fig. 2a and Supplementary Data 6). Unsupervised clustering identified 15 T cell clusters,

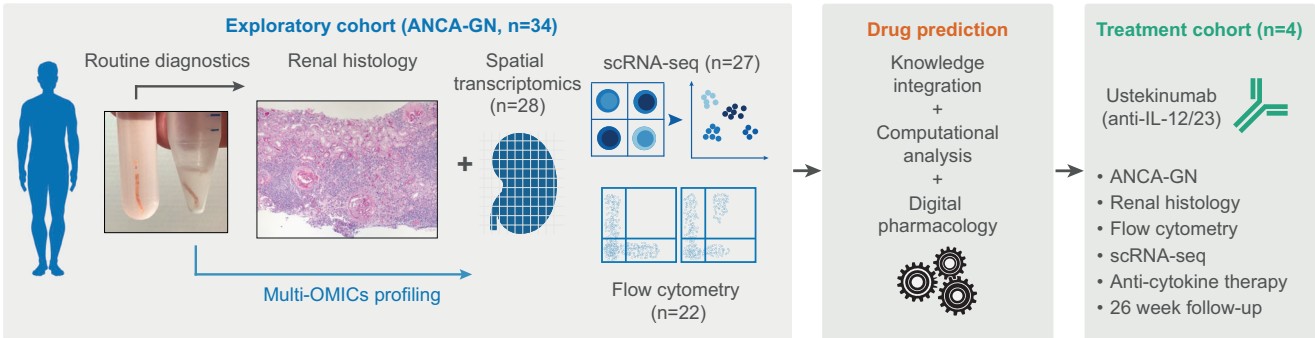

**Fig. 1 | Study overview.** 34 patients from the Hamburg GN Registry underwent diagnostic kidney biopsy and multi-OMIC high dimensional immune profiling. Based on these results drug prediction revealed ustekinumab as the strongest candidate for treatment of ANCA-GN. Subsequently, 4 patients with severely relapsing ANCA-GN were treated with ustekinumab and followed up for 26 weeks.

**Table 1 | Basic and Clinical Characteristics Exploratory cohort**

| | N = 34 |
|---|---|
| **Demographics** | |
| Age—years, median (IQR) | 64.5 (57.75–74.25) |
| Sex, n (%) | |
| Female | 14 (41.18) |
| Male | 20 (58.82) |
| BMI, median (IQR)[a] | 24.65 (22.90–27.36) |
| ANCA status, n (%) | |
| MPO | 22 (64.71) |
| PR3 | 12 (35.29) |
| Initial organ involvement, n (%) | |
| General | 22 (64.71) |
| Renal | 34 (100) |
| ENT | 6 (17.65) |
| Lung (DAH) | 11 (32.35) |
| Nervous system | 3 (8.82) |
| Cutaneous | 2 (5.88) |
| Abdominal | 2 (5.88) |
| Eye | 2 (5.88) |
| Heart | 1 (2.94) |
| Histological ANCA renal risk score[16], n (%) | |
| Low | 12 (35.29) |
| Medium | 15 (44.12) |
| High | 7 (20.59) |
| Laboratory values, median (IQR) | |
| Creatinine (mg/dl) | 2.12 (1.66–4.55) |
| eGFR (ml/min) | 23.5 (11.75–42.75) |
| ACR (mg/g) | 720 (328.5–1500) |
| Immunosuppressant induction treatment, n (%) | |
| Glucocorticoids | 34 (100) |
| Rituximab | 9 (26.47) |
| Cyclophosphamide | 22 (64.71) |
| Cyclophosphamide and Rituximab | 3 (8.82) |
| PLEX | 4 (11.76) |

Source data are provided as a Source Data file.

*IQR* interquartile range, *ANCA* antineutrophile cytoplasmatic antibody, *MPO* myeloperoxidase, *PR3* proteinase 3, *ENT* ear nose throat, *DAH* diffuse alveolar hemorrhage, *PLEX* therapeutic plasma exchange, *eGFR* estimated glomerular filtration rate, *ACR* albumin-creatinine-ratio.
[a]n = 33.

containing CD4$^+$ T effector cells (CD4$^+$ Teff), CD8$^+$ T effector cells (CD8$^+$ Teff), CD4$^+$ naïve T cells, CD8$^+$ naïve T cells, CD8$^+$ T effector memory cells (Teff/em), CD4$^+$ central memory T cells (Tcm), stressed T cells, regulatory T cells (Treg), γδ T cells, mucosal-associated invariant T cells (MAIT), natural killer T cells (NKT), CD4$^+$ cytotoxic T cells (CTL), natural killer cells (NK cells) and proliferating T cells (Fig. 3a and Supplementary Fig. 2b). Cytokine expression analysis revealed the highest cytokine scores in CD4$^+$ and CD8$^+$ T effector cells (clusters 1 and 2) (Fig. 3b and Supplementary Fig. 3a). Interestingly, these effector CD4$^+$ and CD8$^+$ T cells were enriched in the inflamed kidney but not in the peripheral blood, highlighting their relevance in renal inflammation (Fig. 3c). Further analyses showed that the CD4$^+$ T effector cell cluster had a high proportion of Th1, Th1-like, and Th17 cells and CD8$^+$ T effector cell cluster of Tc1, and Tc17-like cells (Fig. 3d and Supplementary Fig. 3b, c). Subgroup analysis of proteinase 3 (PR3) ANCA versus myeloperoxidase (MPO) ANCA patients showed no differences in composition of T effector cells (Supplementary Fig. 3d).

To understand the spatial location of these pathogenic CD4$^+$ and CD8$^+$ effector T cells in the inflamed kidney, we next used single cell information to deconvolve the spatial transcriptomic data. Consistent with the up-regulation of T cell activation markers, CD4$^+$ Th1 and Th17 as well as CD8$^+$ Tc1 cells were exclusively localized to inflammatory glomerular and tubulointerstitial niches (Fig. 3e and Supplementary Fig. 3e).

## Digital pharmacology identifies ustekinumab as drug candidate

Based on the combined analysis of spatial and single cell transcriptome, type 1 and 3 cytokine producing T cells constitute a potential immunopathogenesis-based therapeutic target in ANCA-GN. We employed digital pharmacology, the mapping of drugs to cells based on their molecular interaction, to search for approved drugs that specifically target these pathogenic T cells in the kidney. To narrow our search space to immunomodulating drugs, we preselected 277 drugs consisting of antineoplastic agents, endocrine therapy drugs, and immunosuppressants (Anatomical Therapeutic Chemical (ATC) codes L01, L02, L04, respectively) that could potentially interact with CD4$^+$ and CD8$^+$ Teff subsets in the inflamed glomerular and inflamed tubulointerstitial compartments. To prioritize these drugs, we constructed a dictionary of drug-gene interactions based on the spatial and single cell transcriptome information and subsequently filtered drugs for chemical viability and FDA-approval (Fig. 3f and Supplementary Data 7). Among the drugs with high differential interaction scores in the inflamed renal compartments, we identified ustekinumab as the drug exhibiting the highest specificity for CD4$^+$ and CD8$^+$ effector T cells. Ustekinumab is a human monoclonal antibody directed against the p40 subunit of both IL-12 and IL-23, which has the potential to inhibit Th1/Tc1 and Th17/Tc17 cell responses.

## Rapid biopsy immune profiling

ScRNA-sequencing is a time consuming and expensive technology. To enable rapid and cost-effective screening of pathogenic immune cell infiltrates in a clinical setting, we performed flow cytometry-based single cell immune biopsy profiling of the exploratory cohort (Fig. 4a, b and Supplementary Fig. 4). This approach delivers patient-specific immune profiles within hours after biopsy and might be instrumental in establishing immunopathogenesis-driven targeted biological therapies. Based on our results of the single cell and spatial transcriptomic data, we focused on the identification of pathogenic Th1/Tc1 and Th17/Tc17 cells.

Rapid single cell immune biopsy profiling showed that Th1/Tc1(CXCR3$^+$, CCR6$^-$) and Th17/ Tc17-like cells (CCR6$^+$, CCR4$^+$)[18] were the dominant T cell subsets in the inflamed kidney of ANCA-GN patients (Fig. 4b). Subsequent multiplex immunofluorescence staining showed that these pathogenic T cells were mainly localized to glomerular and interstitial inflammatory areas (Fig. 4c), further supporting an anti-T cell-cytokine treatment with ustekinumab.

## Demographic, clinical, and immune characteristics of the ustekinumab treatment group

Based on our findings from the exploratory ANCA-GN cohort and results from preclinical GN models, we decided to use ustekinumab in combination with low-dose cyclophosphamide in patients with AAV that had relapsing disease and a relative contraindication or incomplete response to current standard therapies (ustekinumab treatment cohort). The basic and clinical characteristics of the treatment group, encompassing four ANCA-GN patients are briefly summarized below and are shown in more detail in Table 2 and Supplementary Fig. 5.

Patient 1: A 73-year-old male, with known MPO-ANCA positive vasculitis under remission maintenance therapy with rituximab, presented with fatigue, dizziness, lower limb edema, gross hematuria and acute kidney injury to our nephrology clinic. Kidney biopsy was performed and revealed active, crescentic ANCA-GN. No other organ manifestation was noted. At time of relapse, the patient received rituximab remission maintenance therapy. Despite appropriate rituximab dosing and intervals, full B cell depletion was not achieved. With known urinary flow obstruction and the need for urinary diversion, the

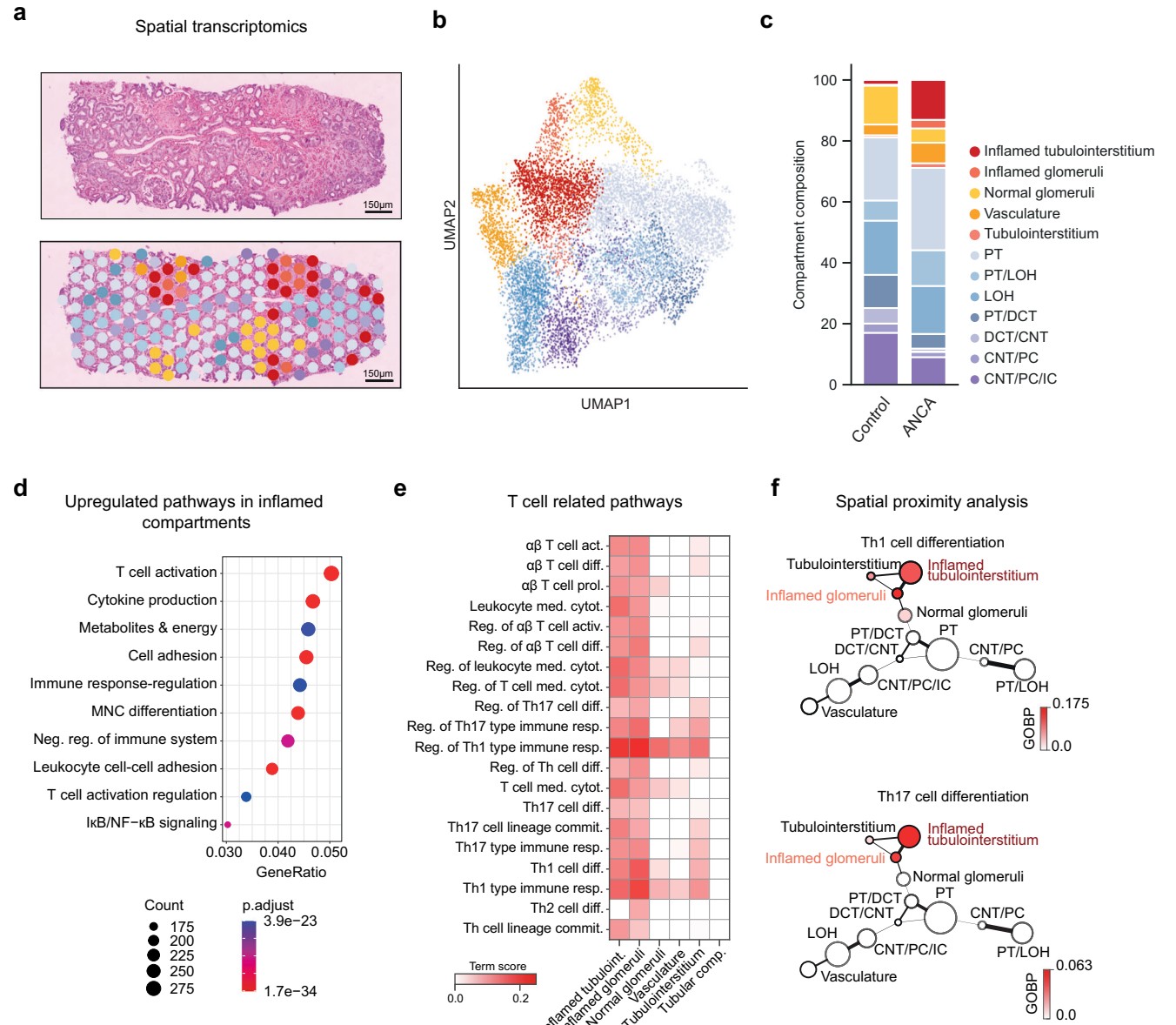

**Fig. 2 | Spatial transcriptome analysis of the ANCA-GN exploratory group.**
**a** Left, Representative section of an H&E-stained kidney biopsy. Right, Spatial distribution of renal compartments overlaid on the representative section. **b** UMAP embedding displaying annotated renal compartments across 10,763 spots from all ST slides. See Supplementary Fig. 1c for the expression of cell type markers in annotated compartments. **c** Barplots showing the composition of renal compartments in the control (21,420 spots) and ANCA-GN exploratory group. The significance of the difference in composition was assessed with differential population analysis and is presented in Supplementary Data 2. **d** Top 10 enriched gene ontology terms in inflamed compartments. Count: number of DE genes in the term. Gene ratios: ratio of the number of DE genes in the term to the number of all DE

genes. The colors show adjusted *p*-values (*p*.adjust) computed using the *enrichGO* function from R package clusterProfiler with right-tailed Fisher's exact test and Benjamini–Hochberg multiple test correction. **e** Scores of alpha-beta T cell-related gene ontology terms computed using GSVA in different renal compartments. **f** Graph showing the spatial proximity of renal compartments and the enrichment of T cell activation. The node sizes and edge widths are proportional to compartment size and spatial proximity, respectively. The nodes are colored by an increasing Th1 and Th17 cell differentiation term scores computed using GSVA. PT, proximal tubules. LOH, loop of Henle. DCT distal convoluted tubules, CNT connecting tubules, PC principal cells, IC intercalated cells. Source data are provided as a Source Data file.

patient was hesitant for full dose cyclophosphamide therapy. (Supplementary Fig. 5a).

Patient 2: A 52-year-old male patient was admitted to our clinic with a creatinine increase as well as active urinary sediment after six pulses of i.v. cyclophosphamide, because of recently diagnosed MPO-ANCA positive vasculitis with extensive organ manifestations. Kidney biopsy was performed and revealed active, crescentic ANCA-GN. The early phase of the COVID-19 pandemic raised substantial concerns over B cell depleting therapies, thus prompted us to decide against re-induction therapy containing rituximab (Supplementary Fig. 5b).

Patient 3: A 32-year-old male was admitted to our nephrology ward with fever, night sweats, weight loss, progressive dyspnea and hemoptysis. Four weeks earlier the patient received rituximab and steroids because of a pulmonary and ENT relapse of known PR3-ANCA positive vasculitis. Chest imaging and bronchoscopy showed progressive DAH (diffuse alveolar hemorrhage). Urinary analysis displayed glomerular hematuria and laboratory testing showed acute kidney injury with pronounced elevation of CRP and ANCA-level. Kidney biopsy was initiated and revealed active, crescentic glomerulonephritis. Because of concomitant severe

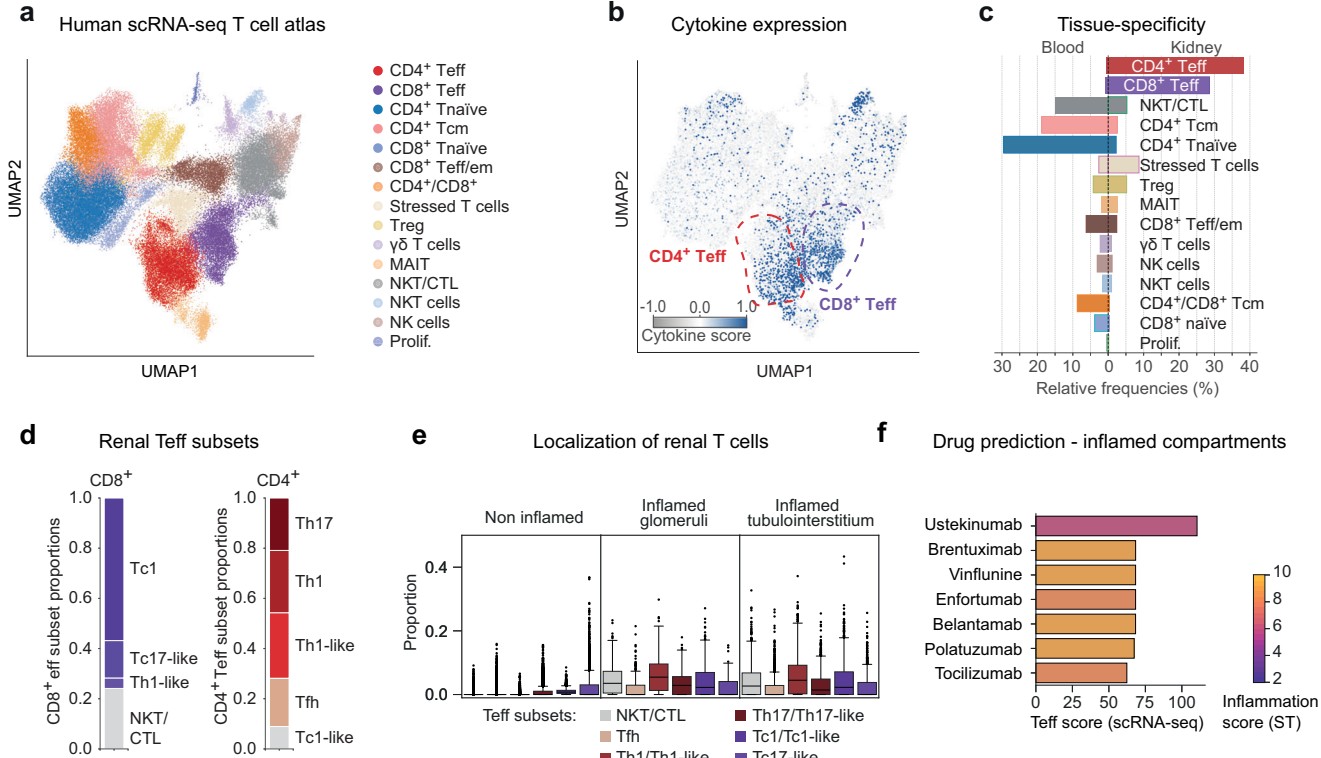

**Fig. 3 | Single T cell transcriptome analysis and drug prediction. a** UMAP projection and cluster annotations of the combined human single cell atlas of renal and blood T cells. In total 72,416 cells are shown of which 22,187 stem from the kidney and 50,229 from the blood. See Supplementary Fig. 2b for the expression of cell type-specific marker genes. **b** Combined type 1-3 cytokine expression score (type 1: *IFNG, TNF, IL2, IL18, LTA, CSF2*; type 2: *IL4, IL5, IL9, IL13*; type 3: *IL17A, IL17F, IL22, IL26*) per cell overlayed on the UMAP projection. The positions of CD4+ and CD8+ T effector cell clusters are highlighted manually. The detailed expression per cytokine type and cell type is shown in Supplementary Fig. 3a. **c** Relative tissue composition per cell type. The frequencies are computed separately for blood and kidney cells, i.e., both sides add up to 100%. **d** CD4+ and CD8+ T effector cell subsets and their relative proportions. Both T effector cell subsets show some overlap due

to the proximity of their respective expression profiles, e.g., the CD4+ Teff cluster contains a small proportion of CD8+ Tc1-like cells. The marker gene expression is detailed in Supplementary Fig. 3b, c. **e** Distribution of Teff cell subsets within the non-inflamed and inflamed compartments. For detailed proportions in individual non-inflamed compartments, see Supplementary Fig. 3e. Boxplots show the median (middle horizontal line), interquartile range (box), Tukey-style whiskers (lines beyond the box), outliers (data points beyond 1.5*interquartile or below−1.5*interquartile) for proportion of Teff subsets in 10,763 spots from all ANCA ST slides. **f** Ranking of drugs based on their interaction scores within Teff single cells, with colors representing their interaction scores specifically within inflamed renal compartments. Source data are provided as a Source Data file.

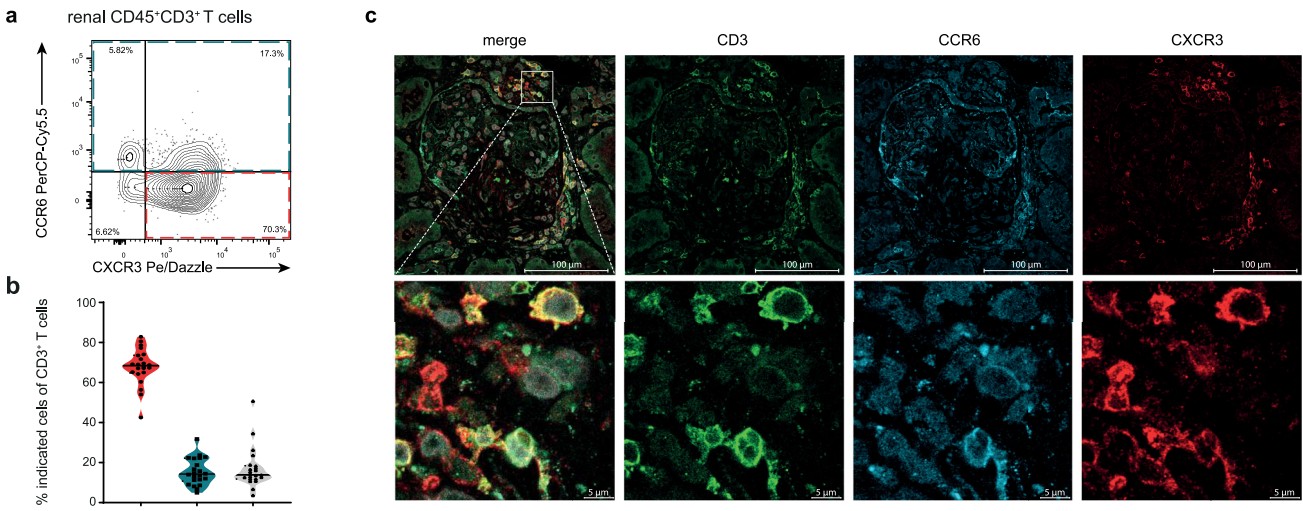

**Fig. 4 | Immune profiling of renal T cells. a** Representative flow cytometry plot showing the identification of chemokine receptor expression from cells isolated from biopsy samples of patients with ANCA-GN (exploratory cohort, *n* = 22). **b** Quantification of chemokine receptor expression CXCR3 (Th1/Tc1) and CCR6 (Th17/Tc17) from renal CD3+ T cells. Violin plots show mean, symbols represent

individual data points. (*n* = 22). **c** Representative immunofluorescence staining of chemokine receptors CXCR3 and CCR6 on CD3+ T cells in human kidney tissue of ANCA-GN. Lower row zoomed in areas. Source data are provided as a Source Data file.

**Table 2 | Basic and Clinical Characteristics treatment cohort**

|  | Patient 1 | Patient 2 | Patient 3 | Patient 4 |
|---|---|---|---|---|
| **Demographics** |  |  |  |  |
| Age—years | 73 | 52 | 32 | 69 |
| Sex (female/male) | M | M | M | F |
| BMI | 34.68 | 25.90 | 24.38 | 27.06 |
| Time until relapse (weeks) | 143 | 16 | 168 | 138 |
| **ANCA status** |  |  |  |  |
| MPO | + | + | − | + |
| PR3 | − | − | + | − |
| **Organ involvement** |  |  |  |  |
| General | + | + | + | + |
| Renal | + | + | + | + |
| ENT | − | + | + | − |
| Lung (DAH) | − | + | + | − |
| Nervous system | − | + | − | − |
| Cutaneous | + | + | − | − |
| Abdominal | − | − | − | − |
| Eye | − | − | − | − |
| Heart | − | + | − | − |
| **Histological ANCA renal risk score[16]** |  |  |  |  |
| Low | − | + | + | − |
| Medium | − | − | − | − |
| High | + | − | − | + |
| **Laboratory values at relapse** |  |  |  |  |
| Creatinine (mg/dl) | 4.25 | 2.27 | 1.88 | 3.42 |
| eGFR (ml/min) | 13 | 32 | 46 | 13 |
| ACR (mg/g) | 3345.7 | 590.0 | 582.3 | 1134.9 |
| **Previous immunosuppressant treatment** |  |  |  |  |
| Glucocorticoids | + | + | + | + |
| Cyclophosphamide | + | + | − | + |
| PLEX | + | − | − | − |
| Rituximab | + | − | + | + |
| Azathioprine | + | − | − | + |

*ANCA* anti-neutrophile cytoplasmatic antibody, *MPO* myeloperoxidase, *PR3* proteinase 3, *ENT* ear nose throat, *DAH* diffuse alveolar hemorrhage, *eGFR* estimated glomerular filtration rate, *ACR* albumin-creatinine ratio, *PLEX* therapeutic plasma exchange.

leukopenia, cyclophosphamide could not be given at full dose (Supplementary Fig. 5c).

Patient 4: A 72-year-old female patient, with known MPO-ANCA positive vasculitis and remission maintenance therapy with rituximab, was sent to our nephrology ward with acute kidney injury and reduced general condition. Chest imaging ruled out relevant thoracic pathologies. Urinary analysis showed glomerular hematuria and kidney biopsy was issued. Here, active crescentic ANCA-GN was seen and diagnosis of relapsing ANCA-GN was made. Because the patient suffered a relapse while being on remission maintenance with rituximab, she was deemed a poor responder to rituximab. Furthermore, she suffered from myelodysplastic syndrome with bicytopenia (leukopenia and anemia), thus full dose cyclophosphamide was deemed unsuitable, because of increased risk for myelotoxicity. (Supplementary Fig. 5d).

Flow cytometry-based rapid immune biopsy profiling in each of the four ANCA-GN patients demonstrated a strong infiltration of Th1/Tc1 and Th17/Tc17-like cells into the inflamed kidney (Supplementary Fig. 6a, b). Additional single cell transcriptome sequencing of the four patients provided a more comprehensive renal T cell profile and confirmed the observed Th1/Tc1 and Th17/Tc17-cell responses (Supplementary Figs. 7a–e, 8a–d and Supplementary Data 8).

## Clinical response of the ustekinumab ANCA-GN treatment group

The four ANCA-GN patients of the treatment cohort were given ustekinumab s.c. (90 mg) in combination with low dose cyclophosphamide and steroids, following the RITUXVAS trial approach[19], as a re-induction therapy. All patients received ustekinumab at weeks 0, 4, 12, and 24 in combination with two to three low doses of cyclophosphamide (cumulative dose 1.5–2.0 g) and glucocorticoids according to the PEXIVAS trial reduced dose regimen[20]. Starting at week 16, patients 3 and 4 (patient 2 at week 22) received a low dose remission maintenance therapy with either azathioprine or mycophenolate mofetil (MMF) (Fig. 5a). At six months, the prednisolone dose was tapered to 5 mg daily in all four patients.

All patients showed a rapid clinical and serological response to this re-induction treatment protocol. Mean serum creatinine levels decreased from a median of 2.8 (2.0–4.0) mg/dl to 1.6 (1.3–1.7) mg/dl at 6 months. According to the albumin-creatinine-ratio, median albuminuria decreased from 862.5 (584–2793) mg/g at the time of relapse to 604 (359–1402) mg/g at 6 months (Fig. 5a). ANCA serum levels decreased from 65 (24–126) U/ml to 32 (9–57) U/ml. The Birmingham vasculitis activity score (BVAS) declined from a median of 12.5 (9.75–13) at the beginning of ustekinumab treatment to 2.5 (2–4.5) at 6 months (Fig. 5b). C-reactive protein (CRP) levels rapidly improved throughout the 6-month treatment period (ranging from 5 to 190 mg/l at the beginning of treatment to < 4–36 mg/l at 6 months) (Fig. 5b). The treatment with ustekinumab was well tolerated. No serious adverse effects were observed during the 6-month treatment period.

## Discussion

Despite numerous advances in therapy for ANCA-GN, these patients still have a substantial risk of kidney failure and increased mortality[21,22]. Today, the leading causes of death in ANCA-GN are infections, followed by cardiovascular disease and malignancies[23], all associated with immunosuppressive therapy. This highlights the unmet need to balance disease control against the risk of side effects. However, the limited understanding of the underlying immunopathology, particularly within the inflamed kidney, impedes the implementation of tailored and effective treatment options. Therefore, in this study we sought to tackle this issue with an approach consisting of four distinct steps.

First, using unsupervised analyses of spatial transcriptomics data derived from the kidneys of patients with ANCA-GN, we identified a distinct enrichment of inflammatory glomerular and tubulointerstitial niches, linked to T cell activation, as the key molecular pathways. Evidence for a pathogenic role of T cells in ANCA-GN patients, is derived from genetic studies showing a significant association with distinct human leukocyte antigen (HLA) class II haplotypes[24–26], an unbalanced activation state of blood and kidney T cells[27–29] and a therapeutic T cell-depletion study in refractory AAV patients[30]. The T cell subsets and cytokine networks that promote tissue injury and loss of renal function, however, remain to be fully elucidated. Thus, secondly, we performed unsupervised single cell transcriptomics and epitope mapping of renal T cells from our exploratory cohort, revealing the dominance of proinflammatory cytokine-producing Th1/Tc1 and Th17/Tc17-like effector T cells in the kidneys of ANCA-GN patients. Thirdly, based on the dominant effector T cell subsets in inflammatory glomerular and tubulointerstitial niches of nephritic kidneys we used digital pharmacology to investigate relevant drugs targeting pathways expressed in these T cells, and could thereby identify ustekinumab as a potential treatment approach. Ustekinumab, a human monoclonal antibody directed against the p40 subunit of both IL-12 and IL-23 thereby targeting the Th1/Tc1 and Th17/Tc17 immune responses, is approved for the treatment of psoriasis[31], psoriatic arthritis[32], and inflammatory bowel disease[33,34]. Several studies highlighted its efficacy and good tolerability in these patients[35–40]. To date, there are no data available for using ustekinumab in ANCA-GN, despite the fact that experimental GN models, including

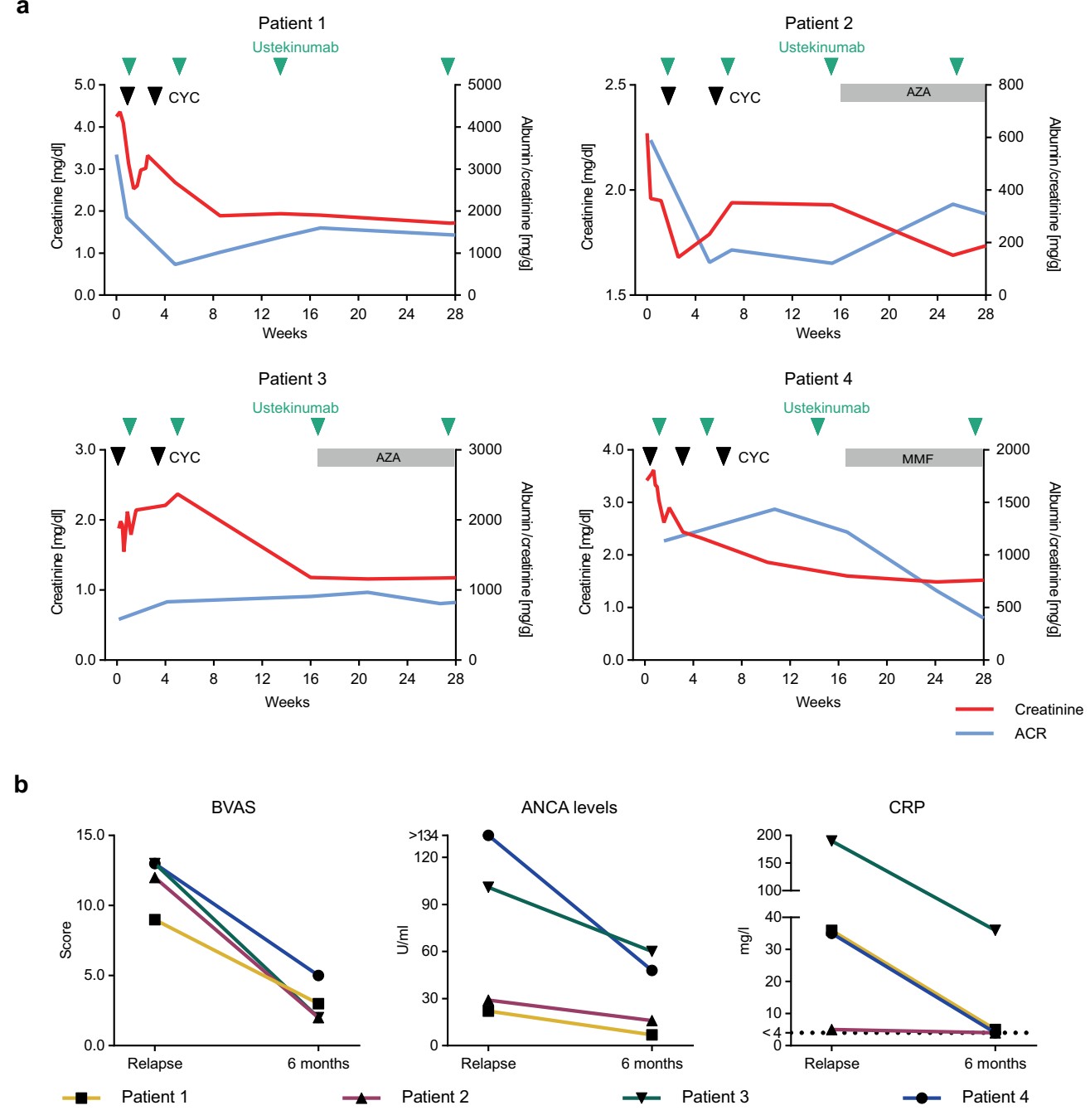

**Fig. 5 | Clinical outcome of the ustekinumab treatment cohort. a** Course of serum creatinine and albuminuria during ustekinumab treatment. Black arrowheads indicate cyclophosphamide and green arrowheads ustekinumab administration. Gray bands indicate low dose remission maintenance therapy with either AZA or MMF. **b** BVAS, ANCA levels measured via ELISA and CRP levels at baseline and 6 months after initiation ustekinumab treatment ($n = 4$). (CYC cyclophosphamide; AZA azathioprine; MMF mycophenolate mofetil; BVAS Birmingham Vasculitis Activity Score; CRP C-reactive protein; ACR albumin creatinine ratio). Source data are provided as a Source Data file.

preclinical ANCA-GN models[41–45], provide a clear rationale for targeting the IL-12/IL-23 axis in immune mediated kidney disease.

Fourthly and finally, given our findings in the exploratory cohort, we assessed the efficacy of ustekinumab in four ANCA-GN patients with relapsing disease. Our treatment protocol was designed as an add-on therapy of ustekinumab at weeks 0, 4, 12, and 24 with up to three low-dose pulses of cyclophosphamide, similar to recent trials establishing rituximab and the complement inhibitor avacopan as add-on treatments for ANCA vasculitis (RITUXVAS and ADVOCATE trials)[8,19]. The immunosuppression was complemented by an intravenous prednisolone and oral glucocorticoid therapy according to the PEXIVAS

study reduced dose regimen[20]. All four patients responded rapidly to therapy, with improvements in kidney function, ANCA levels, CRP, and BVAS. Importantly, all patients tolerated the treatment well and no adverse ustekinumab-related effects were observed.

This report pioneers an immunopathology-based anti-T cell-cytokine therapy for immune-mediated kidney diseases. Our data further suggests that combining high-dimensional single cell immune profiling with clinical and histopathological information facilitates personalized pathogenesis-based treatments. Moreover, this study indicates that rapid single cell immune biopsy profiling by flow cytometry is a feasible approach that might be routinely applied in patients

with ANCA-GN and could prove to be a potential strategy for other organ-specific autoimmune and inflammatory diseases.

Although our results provide an immunopathogenesis-based rationale for targeting Th1/Tc1 and Th17/Tc17 responses with ustekinumab in ANCA-GN, our study has several limitations. The treatment protocol was designed as an add-on therapy, making it more difficult to assess the intrinsic efficacy of ustekinumab, and it is likely that part of the observed response to treatment is due to the concomitant use of low-dose cyclophosphamide and steroids. In addition, our study focused on the renal manifestation of AAV, and it is unclear whether these results also apply to the involvement of other organs.

In addition, long-term data need to be acquired to confirm the overall safety profile of ustekinumab in ANCA-GN. Furthermore, the data generated from this case series is based on a low number of patients without a control group. Therefore, these results should be interpreted with caution and need to be confirmed in adequately designed clinical trials for which the appropriate treatment protocol and patient subgroups remain to be determined. Taken together, our study suggests that ustekinumab is a well-tolerated therapeutic option for the treatment of ANCA-GN, which should be further investigated in clinical trials.

## Methods

### Patients

We included two independent ANCA-GN patient groups from the Hamburg GN Registry[14,16] in our study. The exploratory cohorts consist of 34 patients and the ustekinumab treatment group of four patients. For the spatial transcriptomic analysis of control samples, the healthy parts of the kidney, which was removed due to tumor nephrectomy, were used. Informed consent was obtained from all participating patients in accordance to the CARE guidelines and in accordance with the ethical principles stated in the Declaration of Helsinki. All four patients in the ustekinumab treatment group also provided written informed consent before receiving ustekinumab as an off-label treatment. Detailed information on the patient cohorts and the performed analysis are provided in Tables 1, 2 and Supplementary Data 10.

Sex- and gender-based analyses were not performed. Information about the sex of the patients is provided in Table 1 for the exploratory cohort and in Table 2 for the treatment cohort. Biological sex and self-reported sex were identical in both the exploratory and treatment cohort.

These studies were approved by the Institutional Reviewing Board (IRB) of the University Medical Center Hamburg-Eppendorf and Ethik-Kommission der Ärztekammer Hamburg (local ethics committee of the chamber of physicians in Hamburg), and covered by the licenses PV4806, PV5026, and PV5822.

### Spatial transcriptomics

**Preprocessing of the spatial transcriptomics slides.** For spatial transcriptomics, formalin-fixed paraffin-embedded (FFPE) tissue sections from patients with ANCA-associated glomerulonephritis and controls (healthy tissue from tumornephrectomies) were transferred on Visium (10x Genomics) slides (spatial for FFPE gene expression human transcriptome) and processed according to the manufacturer's instructions. Next-generation sequencing was performed on an Illumina NovaSeq 6000 aiming at 25,000 reads per spot (PE150).

For alignment to the genome of ST slides ($n = 20$) from 30 patients, the human genome assembly GRCh38-2020-A was used. Mapping to the genome was performed using 10x Genomics Space Ranger (v2.0.1). Alignment metrics from spaceranger are provided in Supplementary Data 1. The same alignment method and libraries as used for the exploratory group were used to align ST slides of the internal controls ($n = 3$).

### Quality control

After alignment of the ST slides to the genome, 1 slide was excluded from analysis due to low gene counts (280 median genes per spot compared to $3499.58 \pm 972.97$, Supplementary Data 1). Data analysis of the ST gene expression data was performed using Scanpy[46] (v1.9.3) in Python (v3.9.7). The following parameters in Scanpy's preprocessing pipeline were used to filter poor-quality spots: $min\_genes = 100$, $min\_spots = 3$, $min\_counts = 2000$, $max\_counts = 35000$. The filtered ST data consisted of 10,763 spots and 17,847 genes. The filtered spot counts were normalized to sum to 10,000, and data was $log_2$-transformed with a pseudo-count of 1.

### Clustering and annotation

Principal components ($n\_comps = 50$) were computed on the highly variable genes ($highly\_variable\_genes$ in Scanpy with default settings and slide-name as $batch\_key$). The batch effect corresponding to the slide was removed using harmony[47] (v0.1.0) in R (v4.1.1). To identify clusters, Leiden clustering ($scanpy.tl.leiden$) was performed on Uniform Manifold Approximation and Projection (UMAP) data projections with a resolution of 1.2. The UMAP projections were generated on a neighborhood graph constructed using $scanpy.pp.neighbors$ with $n\_neighbors = 10$. Cluster annotations were performed using the following cell type specific markers from a reference kidney single cell dataset[17] - Proximal tubules (PT): *LRP2, CUBN, SLC13A1*, Distal convoluted tubules (DCT): *SLC12A3, CNNM2, FGF13, KLHL3, LHX1, TRPM6*, Connecting tubules (CNT): *SLC8A1, SCN2A, HSD11B2, CALB1*, Principal cells (PC): *GATA3, AQP2, AQP3*, Intercalated cells (IC): *ATP6VOD2, ATP6V1C2, TMEM213, CLNK*, Ascending thin loop of Henle (Thin limb): *CRYAB, TACSTD2, SLC44A5, KLRG2, COL26A1, BOC*, Thick ascending loop of Henle (TAL): *CASR, SLC12A1, UMOD*. Endothelial cells (Endo): *CD34, PECAM1, PTPRB, MEIS2, EMCN*, vascular smooth muscle cells (vSMC)/Pericyte: *NOTCH3, PDGFRB, ITGA8*, Fibroblasts: *COL1A1, COL1A2, C7, NEGR1, FBLN5, DCN, CDH11*, Podocytes: *PTPRQ, WT1, NTNG1, NPHS1, NPHS2, CLIC5, PODXL*, Immune cells: *PTPRC, CD3D, CD14, CD19*. The expression of these marker genes for each annotated renal compartment is shown in the Supplementary Fig. 1c. The distribution of total gene counts, number of spots across slides, and annotated compartments are presented in the Supplementary Fig. 1a.

### Quantification of spatial proximity

The spatial neighborhood enrichment was performed with Squidpy[48] (v1.2.2) over all slides. The underlying multi-sample spatial graph was constructed by merging all sample-specific spatial graphs into a single graph, resulting in one connected component per sample. The sample-specific graphs were constructed by connecting each spot to its nearest neighbors. To visualize the neighborhood enrichment matrix, the compartments were considered as nodes with the number of spots in a compartment as node sizes, z-scores as edge weights, and *neato* as layout engine from the library Pygraphviz (v1.11). Negative z-scores were set to 0, essentially removing spatially distant compartments. For visualization, the resulting weights were downscaled by 0.25.

### Integration of control samples

ST data from kidney nephrectomies ($n = 8$) was integrated with the previously generated embedding of ST data from the ANCA-GN exploratory group. In total, we used 3 slides generated at the UKE Hamburg (Supplementary Data 9) and 5 slides previously generated by Lake et al. [17]., totaling 21,420 spots. The integration was performed with Symphony[49] (v0.1.0) with highly variable genes computed over the ANCA-GN exploratory group.

### Differential population analysis

To identify the renal compartments differentially abundant between the control and ANCA-GN samples, differential population analysis was applied using scCODA[50] (v0.1.9) with CNT/PC as the reference cell type

and a false discovery rate of 0.05. We identified inflamed glomerular, inflamed tubulointerstitial, PT, PT/LOH, and Tubulointerstitial/Vessels to be differentially abundant between control and ANCA-GN (Supplementary Data 2).

## Gene set enrichment analysis

First, we identified differentially expressed (DE) genes in inflamed glomerular and inflamed interstitial compartments using a Wilcoxon test (adjusted p-value cutoff of 0.05 and log$_2$-fold change cutoff of 0.25) through *scanpy.tl.rank_genes_groups*. We then performed gene set enrichment analysis on a functional level with the differentially expressed (DE) genes as input, using the *enrichGO* function from *clusterProfiler* R package[51] (*v4.2.2*) and *biological processes* as gene ontology (GO). The function *simplify* was used to remove redundant GO terms. Gene-set variation analysis (GSVA)[52] was used to compute the scores of gene set ontology terms.

## Annotation of H&E slides

The same biopsy samples as used in 10x Visium were manually annotated by an expert into three categories: normal, crescentic, and uncertain. The third group contained the tissue that could not be confidently assigned to either normal or crescentic categories. The original images were exported to TIF and processed using ImageJ (v1.54f). The manual annotations were performed using Napari (v0.5.0a2.dev171+gf2d7d437).

## Single cell RNA-sequencing: preprocessing and quality control

The Cell Ranger software (v5.0.1 and v7.1.0, 10x Genomics) was used to demultiplex cellular barcodes and map reads to the reference genome GRCh38-3.0.0 and GRCh38-2020-A. All quality control and preprocessing steps were performed in Seurat[53] (v4.0.4) and R (v4.1.1). The Seurat demultiplexing function *HTODemux* was used to demultiplex the hashtag samples. We removed the cells in which less than 500 or more than 5000 expressed genes were detected. We further filtered out low-quality cells with more than 10% mitochondrial genes. Subsequently, raw counts were normalized to 10,000 and log1p transformed, batch corrected and integrated with harmony[47] using the 2000 most highly variable genes, and clustered using the Louvain algorithm with resolution 0.1. T cells were isolated by removal of all cell clusters with low CD3 expression. We merged the tissue-specific datasets for each cohort by keeping the union of all genes for blood and kidney samples. Subsequently, we removed all cells belonging to the top 0.1% total counts quantile or expressing less than 200 genes as well as any genes that were expressed in less than 10 cells. We further removed cells with more than 12,000 detected surface proteins and proteins that were present in less than 10 cells. In the case of the transcriptome information we normalized the raw counts to sum up to 10,000 and log1p transformed them. For the raw protein counts we performed centered log-ratio normalization. The filtered, processed, and combined single-cell data for the exploratory cohort contains 72,416 cells (22,187 kidneys, 50,229 blood) and 21,419 genes (Supplementary Fig. 2a and Supplementary Data 6). For the treatment group, the combined single-cell data contains 34,810 cells (15,372 kidney, 19,438 blood) and 38,224 genes (Supplementary Fig. 7e and Supplementary Data 8).

## Clustering and cell type identification

In the following, we provide full information only about the analysis workflows for the exploratory group but mention differences to the analogous workflow for the treatment group. All analyses were performed either in R (v4.1.1) using Seurat (v4.0.4) or in Python (v3.9.17) using Scanpy[46] (v1.9.1).

For the exploratory group, we combined two integration workflows to identify and annotate cell types: one to identify broad T cell clusters and another to annotate specific CD4$^+$ and CD8$^+$ T cell subsets. We first performed a principal component analysis on the top 2000

highly variable genes and then applied harmony[47] on the first 30 principal components to correct batch effects between patients. After computing the nearest neighbor graph, we clustered the data using the Louvain algorithm with a resolution of 0.1. We annotated the cell clusters using canonical cell type markers for broad T cell subsets (Fig. 3a and Supplementary Fig. 2b). Type 1-3 cytokine scores (type 1: *IFNG, TNF, IL2, IL18, LTA, CSF2*; type 2: *IL4, IL5, IL9, IL13*; type 3: *IL17A, IL17F, IL22, IL26*) were calculated using the Scanpy function *score_genes* (Fig. 3b and Supplementary Fig. 3a).

In a second step, we isolated the CD4$^+$ Teff and CD8$^+$ Teff cells, mostly composed of kidney cells, and removed patients containing less than 2 total cells (blood and kidney) and genes present in less than 10 cells. We reintegrated the remaining cells with totalVI[54] based on the top 4000 highly variable genes and all surface proteins, treating the patient ID as a categorical covariate. After Leiden clustering with a resolution of 0.8 and 1.0, respectively, we annotated the cell clusters based on canonical markers for CD4$^+$ and CD8$^+$ T cell subsets (Fig. 3d and Supplementary Figs. 2b and 3b,c).

For the ustekinumab group, we followed an analogous integration, clustering, and annotation workflow (Supplementary Figs. 7a–e and 8a–d) with the following differences. First, we regressed out the counts based on the number of genes, the total number of counts, and the fraction of mitochondrial genes per cell before integration of the full dataset and the T effector subsets. Secondly, we performed the reintegration of both T effector subsets with harmony instead of totalVI.

## Cell type deconvolution of spatial transcriptomics

In combination with the ANCA-GN renal T cell single cell atlas described in this study, the single cell atlas from Stewart et al. [55]. was integrated to estimate cell type proportions, resulting in total 24 cell types: LOH, B cell, CD, CD4$^+$ T naive, CD4$^+$ Tcm, CD8$^+$ T naive, CNT, DC, Endo, Fib, Macrophage, Mast cell, Monocyte, Myofib, Neutrophil, NKT, Podo, PT, Tfh, Treg, Th1 and Th17, Tc1 and Tc17. Cell-type deconvolution was performed using a reference-based algorithm, DISSECT[56], with default training parameters. To generate simulated data from training using the PropsSimulator module of DISSECT, the following changes were made: *n_samples* = 8000 and *downsample* = 0.1.

## Drug prediction

Drugs from ATC/DDD classification L (Antineoplastic and immunomodulating drugs), excluding immunostimulants from ATC/DDD classification L03, were used as potential candidate drugs. The targets of these drugs were extracted from ChEMBL[57] and putative drug interactions from the drug-target interaction database DGIdb[58], resulting in a total of 277 drugs. To assess the targets of the drugs, we used drug2cell (v0.1.0)[59] with inflamed glomerular and interstitium renal compartments as clusters of interest. Default parameters were used in drug2cell in addition to a cutoff of 0.25 for log$_2$-fold change in the drug-target expression. This resulted in 14 drugs (Supplementary Data 7) whose targets were differentially enriched in the two compartments. To prioritize drugs, we looked for drugs that affect primarily the inflamed compartments and are less enriched in others. Only drugs that were enriched at least 75% in the compartments of interest and less than 75% in all other compartments were selected. This criterion resulted in 7 potential drugs targeting the inflamed compartments: belantamab mafodotin, brentuximab vedotin, vinflunine, ustekinumab, enfortumab vedotin, tocilizumab, and polatuzumab vedotin. To select the final target out of these drugs and to integrate T cell specificities of the drugs, we computed scores of their targets in the CD4$^+$ Teff and CD8$^+$ Teff clusters using *scanpy.tl.score_genes* function and ordered them in decreasing order.

## Isolation and flow cytometry of human biopsy leukocytes

Single-cell suspensions were obtained from human kidney biopsies by enzymatic digestion in RPMI 1640 medium with collagenase D at 0.4 mg/ml (Roche, 11088858001) and deoxyribonuclease I (DNase I;

10 µg/ml; Sigma-Aldrich, 10104159001) at 37 °C for 30 min followed by dissociation with gentle MACS (Miltenyi Biotec). Leukocytes from blood samples were separated using Leucosep tubes (Greiner Bio-One, 10349081). Cells were stained with fluorochrome-conjugated antibodies from BioLegend and BD Biosciences, CD45 BV510 (BioLegend, clone HI30, catalog number 304036, dilution 1:100), CD3 BV785 (BioLegend, clone OKT3, catalog number 317330, dilution 1:200), CD4 BV650 (BioLegend, clone RPA-T4, catalog number 300536, dilution 1:200), CD8 APC-R700 (BD Biscoences, clone RPA-T8, catalog number 565165, dilution 1:100), CXCR3 Pe/Dazzle (BioLegend, clone G025H7, catalog number 353736, dilution 1:100), CCR6 PerCP-Cy5-5 (BioLegend, clone G034E3, catalog number 353406, dilution 1:100). Cells were also stained with a dead cell stain (Molecular Probes, L10119) to exclude dead cells from analysis. Electronic compensation was performed with antibody (Ab) capture beads stained separately with individual monoclonal antibodies (MABs) used in the experimental panel. FACS was performed on a FACSAria Fusion cell sorter (BD Biosciences). Data analysis was performed using FlowJo software (Treestar) or FACSDiva software (BD Biosciences).

### FACS and scRNA-seq processing of human leukocytes
Single-cell suspension of human leukocytes was prepared as described in the section Isolation and Flow cytometry of human biopsy leukocytes. ScRNA-seq of human samples from the kidney and peripheral blood was performed from FACS-sorted CD3 positive T cells using the Chromium Next GEM Single Cell 5′ Kit v2 (10x Genomics) according to manufacturer's instructions. The gating strategy, shown in Supplementary Fig. 4, is identical for the flow cytometry analysis as well as the FACS sorting. It is based on leukocytes, singlets, the living cells, CD45, and for sorting the CD3 population was collected. Libraries were sequenced aiming at 50,000 reads per cell on an Illumina NovaSeq (P150), using the CG00330 protocol from 10X Genomics.

### Immunofluorescence staining
For immunofluorescence staining, paraffin-embedded kidney sections (2 µm) from ANCA-GN patients were stained with primary antibodies against CD3 (Abcam, ab11089, dilution 1:100), CCR6 (Sigma, HPA014488/Origene TA316610, dilution 1:100), and CXCR3 (BD Biosciences, 557183, dilution 1:100) after dewaxing and antigen retrieval (pH6 for 15 min). Following washing in phosphate-buffered saline, fluorochrome-labeled secondary antibodies were applied. Staining was visualized using an LSM800 with Airyscan and the ZenBlue software (all Carl Zeiss, Jena, Germany).

### Reporting summary
Further information on research design is available in the Nature Portfolio Reporting Summary linked to this article.

## Data availability
All gene expression data used in this manuscript are publicly available via the NCBI Gene Expression Omnibus (https://www.ncbi.nlm.nih.gov/geo/). The newly generated data for this study is accessible under GSE253633 and GSE250138. The accession codes for all other gene expression data used are listed in the Supplementary Data 10. Source data are provided with this paper.

## Code availability
The code to process and analyze the single cell sequencing and ST data is available at https://github.com/imsb-uke/ANCA-GN_transcriptomics. The source code is also deposited at Zenodo (https://doi.org/10.5281/zenodo.13208437).

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

## Acknowledgements

This study was supported by grants from the Deutsche Forschungsgemeinschaft (DFG) to U.P. (SFB 1192 A1 and C3), C.F.K. (SFB 1192 A5 and C3; KR 3483/3-1) and S.B. (SFB 1192 A2, B8, and C3). R.K. was additionally supported by the 3R (Replace, Reduce, Refine) funding of the UKE. FACS was performed at the UKE FACS sorting core facility. Single-cell RNA sequencing was performed at the UKE Single Cell Core Facility.

## Author contributions

Conceptualization: S.B., C.F.K., and U.P. Methodology: J.E., R.K., D.P.S., Y.Z., H.J.P., Z.S., N.A., Formal analysis: J.E., R.K., D.P.S., Y.Z., H.J.P., N.A., S.B., C.F.K., and U.P. Spatial transcriptomics: R.K., D.P.S., Z.S., V.S., A.P., A.K., S.L.J.S., T.G.S., V.G.P., T.W., S.B., C.F.K., and U.P., scRNA sequencing: R.K., D.P.S., Y.Z., and C.F.K. Data analysis: J.E., R.K., D.P.S., Y.Z., H.J.P., Renal histology: T.W., and U.P. Patient cohorts: J.E., J.H.R., U.O.W., O.M.S., E.H., T.W., T.B.H., C.F.K., and U.P. Writing original draft: S.B., C.F.K., and U.P. Writing review and editing: J.E., R.K., D.P.S., Y.Z., J.H.R, S.B., J.E.T., H.W.M., C.F.K., and U.P. Visualization: J.E., R.K., D.P.S., S.B., C.F.K., and U.P. Supervision: S.B., C.F.K. and U.P. Funding acquisition: T.B.H., S.B., C.F.K., U.P.

## Funding

## Competing interests

The authors declare no competing interests.
