## [Peer Review File · Nature Communications]

Immune profiling-based targeting of pathogenic T cells with ustekinumab in ANCA-associated glomerulonephritisEditorial Note: This manuscript has been previously reviewed at another journal. This document only contains reviewer comments and rebuttal letters for versions considered at Nature Communications.

REVIEWER COMMENTS

Reviewer #1:

[Note from editor: Reviewer 1 was no longer available for review. Reviewer #3 was therefore asked to assess the authors' responses to the comments raised by this reviewer.]

Reviewer #2 (Remarks to the Author):

The authors have answered adequately in most of my previous points.

Reviewer #3 (Remarks to the Author):

The authors have adequately addressed my comments and concerns in the revised version of this manuscript originally submitted to Nature Medicine.

I have no further requests.

I have also carefully reviewed the comments and concerns raised by Reviewer 1 and the responses from the authors to them.

Even though I do not feel comfortable speaking for Reviewer 1, it is my opinion that the authors have appropriately responded to Reviewer 1's comments, made changes where necessary, and contradicted some concerns appropriately.

Reviewer #4 (Remarks to the Author):

In this study by Engesser et. al, the authors evaluate the transcriptomic signature of ANCA-vasculitis. The investigators use a combination of scRNA-seq, Visium ST, flow cytometry, and histology to identify inflammatory gene expression patterns. They identify Ustekinumab, as an off label indication, for ANCA therapy and select a small cohort of 4 individuals to receive it.

This submission is a revision. The authors were reasonably responsive to the prior reviewers' concerns.

I reviewed the investigators' informatics approach. While certain stylistic differences are present, the approach falls within the spectrum of appropriate and is generally rigorous - on a conceptual level. The datasets are appropriately normalized and clustered. The QC data is acceptable. The approach of co-clustering using orthogonal datasets (in this case scRNA-seq and ST) is becoming more common place and I believe it to be valid. I examined the github page and found the files to be transparent and accessible - for what was there.

However, a critical issue is that I was not able to download any files to reproduce analyses. I

would suggest the authors make all FASTQ, BAM, Tiff, Loupe, and metadata files available to both the reviewers and the scientific community for the ST and scRNA-seq. The `seurat` object (or equivalent) should be included. I did find a GSE number for certain samples in the supplemental Table 10, but the GSE files were blocked and no token was provided. Further, not all samples had a GSE number. Thus, I could not reproduce the investigators' analysis or apply the code from their `.ipynb` file.

The authors did not exclude edge spots, which is a reasonably common practice due to edge artifact signature changes. The authors also did not annotate sclerotic non-inflammatory glomeruli which would have likely clustered separately than the healthy glomeruli or inflammatory glomeruli. Both of these changes would require restarting the analysis from scratch and I doubt either change would have a substantial impact on the results, so I am not requesting it provided the code and analysis from the raw files are otherwise reproducible. Another way to justify not reanalyzing is that the effect size of the inflammatory signature was large enough that the study did not require removal of edge spots or elimination of sclerotic glomeruli to identify the signal.

In summary, this study is a complete story without major flaws in approach, but requires a reproducibility check of the raw data and code. I reviewed the prior reviewers' critiques, which were quite valid, but do not temper my enthusiasm for this study. For example, I understand that prior etanercept studies were negative and that Ustekinumab is an unproven therapy, but the authors have now reduced the strength of their conclusions. Regardless of whether Ustekinumab pans out as a therapeutic option, the authors raise awareness of the possibility for future trials. There is value beyond the Ustekinumab portion of the study because the OMICs datasets hold great value and can be mined for additional drug targets going forward.

Comments to the reviewers

Reviewer #1:

Reviewer 1 was no longer available for review.

Reviewer #2:

The authors have answered adequately in most of my previous points.

Reply: Thank you very much for this comment.

Reviewer #3:

The authors have adequately addressed my comments and concerns in the revised version of this manuscript originally submitted to Nature Medicine. I have no further requests.

Reply: Thank you very much for the supportive comment.

I have also carefully reviewed the comments and concerns raised by Reviewer 1 and the responses from the authors to them. Even though I do not feel comfortable speaking for Reviewer 1, it is my opinion that the authors have appropriately responded to Reviewer 1's comments, made changes where necessary, and contradicted some concerns appropriately.

Reply: We would like to thank the reviewer for the positive and helpful comments

Reviewer #4:

In this study by Engesser et. al, the authors evaluate the transcriptomic signature of ANCA- vasculitis. The investigators use a combination of scRNA-seq, Visium ST, flow cytometry, and histology to identify inflammatory gene expression patterns. They identify Ustekinumab, as an off label indication, for ANCA therapy and select a small cohort of 4 individuals to receive it. This submission is a revision. The authors were reasonably responsive to the prior reviewers' concerns.

Reply: Thank you very much for the supportive comment.

I reviewed the investigators' informatics approach. While certain stylistic differences are present, the approach falls within the spectrum of appropriate and is generally rigorous - on a conceptual level. The datasets are appropriately normalized and clustered. The QC data is acceptable. The approach of co-clustering using orthogonal datasets (in this case scRNA-seq and ST) is becoming more common place and I believe it to be valid. I examined the github page and found the files to be transparent and accessible - for what was there.

Reply: Thanks.

However, a critical issue is that I was not able to download any files to reproduce analyses. I would suggest the authors make all FASTQ, BAM, Tiff, Loupe, and metadata files available to both the reviewers and the scientific community for the ST and scRNA-seq. The seurat object (or equivalent) should be included. I did find a GSE number for certain samples in the supplemental Table 10, but the GSE files were blocked and no token was provided. Further, not all samples had a GSE number. Thus, I could not reproduce the investigators' analysis or apply the code from their .ipynb file.

Reply: We apologize for not providing the access token earlier. The FASTQ files and other files required for alignment have been deposited at the Sequence Read Archive (SRA) through Gene Expression Omnibus (GEO) and are now made publicly available. The ST data and Scanpy h5ad objects can be accessed at <https://www.ncbi.nlm.nih.gov/geo/query/acc.cgi?acc=GSE250138>.

The updated Supplemental Table 10 now contains all links for the full access to the spatial and single cell transcriptomics data.

The authors did not exclude edge spots, which is a reasonably common practice due to edge artifact signature changes.

Reply: We thank the reviewer for these comments. We agree that edge spots could potentially have artifacts and due to the partial tissue coverage, they may have low quality. In our analysis, we have excluded the spots that have a total count less than 2000 and total genes expressed less than 100. This resulted in removal of 1468 spots from 12444 spots in the raw data. We believe that this is a sound approach to remove low quality spots. Further, we have checked whether the edge spots differ significantly from spots in between tissue in terms of total expressed genes and total counts. We did not observe a systemic issue between the edge and within tissue spots. Further, the edge spots are distributed across compartments and they do not form a cluster.

Figure legend (for reviewer only): UMAP of all spots from ANCA-GN ST data (blue: false/spot not on edge; orange: true/spot on edge).

Figure legend (for reviewer only): Boxplots showing total count per spot (blue: false/spot not on edge; orange: true/spot on edge).

The authors also did not annotate sclerotic non-inflammatory glomeruli which would have likely clustered separately than the healthy glomeruli or inflammatory glomeruli.

Reply: In a broad clustering, we do not see a separation on spots within sclerotic glomeruli and inflamed glomeruli. Following your comment, we performed sub-clustering of spots coming from

glomeruli and could identify a subset of spots that show only fibrosis without a high presence of immune cells. These spots differ in their profile from the normal and inflamed glomeruli.

Figure legend (for reviewer only): Expression of podocyte membrane proteins (NPHS1, NPHS2, PODXL), selected immune cell type markers (PTPRC, LYZ, CD3) and fibrosis markers (COL1A1, FN1, VIM, CCN2) in sub-clusters of spots coming from glomeruli.

Excluding these sclerotic spots, we performed gene set enrichment of inflamed compartments again and did not observe a difference in the top gene ontology terms in comparison to manuscript Figure 2d. A comparison of inflamed glomerular with normal glomerular also revealed T cell activation as a prominent pathway.

Figure legend (for reviewer only): Top 10 enriched gene ontology terms in inflamed compartments and inflamed versus healthy glomerular compartment.

Both of these changes would require restarting the analysis from scratch and I doubt either change would have a substantial impact on the results, so I am not requesting it provided the code and analysis from the raw files are otherwise reproducible. Another way to justify not reanalyzing is that the effect size of the inflammatory signature was large enough that the study did not require removal of edge spots or elimination of sclerotic glomeruli to identify the signal.

Reply: We thank the reviewer for the statement that these changes likely do not have an impact on the results. Indeed, our reanalysis, as presented in our previous answers, shows that excluding edge spots and separating sclerotic glomeruli did not have a substantial impact on the results. Therefore, we have kept the analysis in the manuscript as it is.

In summary, this study is a complete story without major flaws in approach, but requires a reproducibility check of the raw data and code. I reviewed the prior reviewers' critiques, which were quite valid, but do not temper my enthusiasm for this study. For example, I understand that prior etanercept studies were negative and that Ustekinumab is an unproven therapy, but the authors have now reduced the strength of their conclusions. Regardless of whether Ustekinumab pans out as a therapeutic option, the authors raise awareness of the possibility for future trials. There is value beyond the Ustekinumab portion of the study because the OMICs datasets hold great value and can be mined for additional drug targets going forward.

Reply: We thank the reviewer for summarizing and noting the significance of this work.

REVIEWERS' COMMENTS

Reviewer #4 (Remarks to the Author):

I am satisfied with the author response.

Reviewer #4 (Remarks on code availability):

The code is adequate, the data was accessible.

Point-by-point response to the reviewer comments

REVIEWERS' COMMENTS

Reviewer #4 (Remarks to the Author):

I am satisfied with the author response.

Reply: We are happy to hear that we could address all the points raised by the reviewer.

Reviewer #4 (Remarks on code availability):

The code is adequate, the data was accessible.

Reply: Thank you!